# Stasis Leg Ulcers: Venous System Revises by Triggered Angiography Non-Contrast-Enhanced Sequence Magnetic Resonance Imaging

**DOI:** 10.3390/diagnostics10090707

**Published:** 2020-09-17

**Authors:** Chien-Wei Chen, Yuan-Hsi Tseng, Min Yi Wong, Chao-Ming Wu, Bor-Shyh Lin, Yao-Kuang Huang

**Affiliations:** 1Department of Diagnostic Radiology, Chang Gung Memorial Hospital, Chang-Gung University, Chiayi 61363, Taiwan; chienwei33@gmail.com (C.-W.C.); mynyy001@gmail.com (M.Y.W.); 2Department of Cardiovascular Surgery, Wound and Vascular Center, 6 West Section, Chang Gung Memorial Hospital, Chia-Yi 61363, Taiwan; 8802003@cgmh.org.tw (Y.-H.T.); jerming@cgmh.org.tw (C.-M.W.); 3Division of Thoracic and Cardiovascular Surgery, Chiayi Chang Gung Memorial Hospital, College of Medicine Chia-Yi and Chang Gung University, Taoyuan 61363, Taiwan; 4Institute of Imaging and Biomedical Photonics, National Chiao Tung University, Tainan 71150, Taiwan; borshyhlin@gmail.com

**Keywords:** MRI, non-contrast, phase contrast, venography, TRANCE, stasis ulcer, chronic wound

## Abstract

Objectives: The distribution of venous pathology in stasis leg ulcers is unclear. The main reason for this uncertainty is the lack of objective diagnostic tools. To fill this gap, we assessed the effectiveness of triggered angiography non-contrast-enhanced (TRANCE)-magnetic resonance imaging (MRI) in determining the venous status of patients with stasis leg ulcers. Methods: This prospective observational study included the data of 23 patients with stasis leg ulcers who underwent TRANCE-MRI between April 2017 and May 2020; the data were retrospectively analyzed. TRANCE MRI utilizes differences in vascular signal intensity during the cardiac cycle for subsequent image subtraction, providing not only a venogram but also an arteriogram without the use of contrast agents or radiation. Results: TRANCE MRI revealed that the stasis leg ulcers of nine of the 23 patients could be attributed to valvular insufficiency and venous occlusion (including deep venous thrombosis [DVT], May–Thurner syndrome, and other external compression). Moreover, TRANCE MRI demonstrated no venous pathology in five patients (21.7%). We analyzed TRANCE MRI hemodynamic parameters, namely stroke volume, forward flow volume, backward flow volume, regurgitant fraction, absolute volume, mean flux, stroke distance, and mean velocity, in the external iliac vein, femoral vein, popliteal vein, and great saphenous vein (GSV) in three of the patients with valvular insufficiency and three of those with venous occlusion. We found that the mean velocity and stroke volume in the GSV was higher than that in the popliteal vein in all patients with venous valvular insufficiency. Conclusions: Stasis leg ulcers may have no underlying venous disease and could be confirmed by TRANCE-MRI. TRANCE MRI has good Interrater reliability between Duplex study in greater saphenous venous insufficiency. It also potentially surpasses existing diagnostic modalities in terms of distinguishable hemodynamic figures. Accordingly, TRANCE-MRI is a safe and useful tool for examining stasis leg ulcers and is extensively applied currently.

## 1. Background

Stasis leg ulcers (or venous leg ulcers, formally) are chronic wounds that are characterized by an irregular shape with well-defined borders and are typically located in the distal calf and perimalleolar region (“gaiter” area). The wounds are usually limited to the subcutaneous plane; however, secondary infection can destroy deep soft tissue. By contrast, leg wounds due to arterial occlusive disease are characterized by rapid progression with a high incidence of limb loss [1,2,3].

Stasis leg ulcers are caused by previous ambulatory venous hypertension of the lower extremity. The severity of ambulatory venous hypertension varies; high severity may be caused by primary venous valvular reflux, right-sided heart failure, poor lymph drainage, and deep vein thrombosis (DVT) with multilevel valve reflux alone or concomitant with venous outflow obstruction. Stasis leg ulcers are also associated with venous occlusion in the pelvic veins, which is not easily detected by duplex or computed tomographic (CT) venography.

The triggered angiography non-contrast-enhanced (TRANCE) technique records differences in vascular signal intensity during the cardiac cycle for subsequent image subtraction [4]. It provides both venograms and arteriograms without the use of contrast agents and currently serves as an alternative in the evaluation of possible complicated vascular status in the Wound and Vascular Center at our institution [5,6]. In this study, we present our experience of applying TRANCE-magnetic resonance imaging (MRI) for evaluating the venous status of patients with stasis leg ulcers.

## 2. Materials and Methods

### 2.1. Patients

The Institutional Review Board (IRB) of Chang Gung Memorial Hospital approved this study (IRB number: 201802137B0 and IRB 201901058B0). The study involved consecutive patients who had been evaluated using TRANCE-MRI for venous pathology in their lower extremities at a tertiary hospital between April 2017 and May 2020. We prospectively collected and retrospectively analyzed their data to determine their clinical significance. All patients who were suspected of having complex venous pathology in their lower extremities were included initially. Patients were excluded if they were pregnant or had non-MRI-compatible ferromagnetic implants. In addition, patients were excluded if they exhibited poor compliance or had multiple comorbidities that prevented them from lying down for the 1-h TRANCE-MRI protocol. Initially, 104 patients were evaluated. One patient was excluded because of possible pregnancy, and three additional patients were either morbidly obese or restless, rendering them unable to continue with the MRI study. Among the remaining 100 patients, 23 received further survey for their stasis leg ulcers in the Wound and Vascular Center at our institution (Figure 1).

All 23 patients underwent noninvasive color Doppler ultrasonography (US) to evaluate the venous status of their lower extremities before the scheduled TRANCE-MRI. The Doppler examinations were performed in the supine position. The femoral vein, great saphenous vein (GSV), popliteal vein, and perforating vein in the calves were examined. Pelvic veins were not evaluated in the Doppler exams.

### 2.2. MRI Acquisition

MRI is performed using a 1.5 T MRI scanner (Philips Ingenia, Philips Healthcare, Best, The Netherlands). The imaging process is performed with the patients in the supine position; a peripheral pulse unit trigger is used for imaging (Figure 2). All arterial system images are evaluated through a three-dimensional (3D) turbo spin-echo (TSE) technique during systole and diastole periods. TSE TRANCE imaging is executed using the following parameters: repetition time (TR), one beat; echo time (TE), shortest; flip angle, 90°; voxel size, 1.7 × 1.7 × 3 mm^3^; field of view (FOV), 350 × 420. During systole, arterial blood flow is relatively fast, which causes signal dephasing and leads to flow voids. Accordingly, when systolic triggering is applied, the arteries would appear black. During diastole, blood flow in the arteries is slow; the signal would not be dephased. Hence, the arteries would appear bright on diastolic scans. Subtracting the two phased scans yields a 3D data set of the arteries only. Other images of the venous systems are evaluated through 3D TSE short-tau inversion recovery (STIR) during the systole period. TSE STIR TRANCE imaging is executed using the following parameters: TR, 1 beat; TE, 85; inversion recovery delay time, 160; voxel size, 1.7 × 1.7 × 4 mm^3^; FOV, 360 × 320. STIR provides extra background suppression because fat and bones are also suppressed. When systolic triggering is applied, the arteries would appear black. The imaging process yields a 3D data set of the venous system, and no subtraction is required for the data set. A quantitative flow scan is routinely performed to determine the appropriate trigger delay times for systolic and diastolic triggering. All images are acquired without the use of gadolinium contrast medium. The TRANCE-MRI protocol requires 40 min for image acquisition, 25 min for magnetic resonance venography (MRV, Appendix A), and 15 min for magnetic resonance arteriography (MRA, Appendix A).

## 3. Results

This study included 23 patients, and the patients’ demographic and medical data regarding gender, age, substance use, comorbidities, history of venous surgery, and venous interventions after TRANCE-MR examination are summarized in Table 1. The most frequent commodities were hypertension and diabetes mellitus. Four patients had malignant disease.

Each of the 23 patients underwent a duplex ultrasound scan and TRANCE-MRI for the venous system (Table 2). Nine patients were observed to have venous junction insufficiency after survey (prominently in the GSV, along with an inverse GSV/PV stroke volume ratio in TRANCE-MRI vs. saphenous-femoral junction reflux in duplex ultrasound). Six patients had DVT, three of whom were identified through duplex ultrasound and the other three were detected through TRANCE-MRI. Furthermore, TRANCE-MRI revealed that one patient had a double inferior vena cava, one patient had situs inversus, and four patients had arterial compression of the iliac veins (May–Thurner [MT] syndrome). Duplex ultrasound examination and TRANCE-MRI revealed no venous lesions in seven and five patients, respectively.

We classified the 23 patients into three groups according to the TRANCE-MRI evaluations and analyzed their wound status and interventions (Table 3). The stasis leg ulcers observed in nine patients were attributed to valvular insufficiency, and those observed in nine patients were attributed to venous occlusion (including DVT, MT syndrome, and other external compression). TRANCE-MRI revealed no venous pathology in five patients (21.7%). Three patients were lymphedema related to radiation and chronic trauma. Two patients were found with coexisted vasculitis. Interrater reliability of Cohen’s Kappa coefficient for examining the ability to detect prominent greater saphenous venous insufficiency between TRANCE-MRI and venous duplex was 0.7315, indicated that there is substantial agreement between the two imaging modalities. (sensitivity 80%, specificity 92.3%, positive predictive value 88.9%, negative predictive value 85.7%). The leg ulcers attributed to venous valvular insufficiency were often located in the gaiter area and were single wounds that involved only one leg. The ulcers attributed to venous occlusion often involved the proximal calf and sole and were multiple wounds that involved both legs. The stasis leg ulcers without any venous pathology, identified by TRANCE-MRI, were often complicated and deep. All patients with venous occlusion received oral anticoagulation therapy.

Hemodynamic parameters of TRANCE-MRI, comprising stroke volume, forward flow volume, backward flow volume, regurgitant fraction, absolute volume, mean flux, stroke distance, and mean velocity, were analyzed in the external iliac vein, femoral vein, popliteal vein, and GSV in three patients with valvular insufficiency and three patients with venous occlusion. Paradoxically, we observed that the mean velocity and stroke volume in the GSV were higher than those in the popliteal vein in all patients with valvular insufficiency (Figure 3).

## 4. Discussion

Venous leg ulcers, or stasis leg ulcers are characterized by an irregular shape with well-defined borders and are typically located in the perimalleolar region (“gaiter” area). The main etiology of such ulcers was previously identified to be ambulatory venous hypertension, which may be caused by primary venous valvular reflux, right-sided heart failure, poor lymph drainage, DVT, and venous outflow obstruction. Other pathophysiology of venous return and of venous ulcer development is proposed, including inadequate microorganisms’ colonization, iceberg hypothesis of chronic venous insufficiency, and venous hypoxia in the microcirculation in the skin level [1,7,8,9]. However, other etiologies of unhealed leg ulcers are not easily differentiated according to conventional venous evaluation [1,3,10].

Most patients with stasis leg ulcers undergo US at the beginning of their therapy. US is operator-dependent, time consuming, and inadequate for providing information regarding pelvic and abdominal areas [11]. Notably, in US, leg ulcers usually obstruct the view for tracking perforating veins. Conventional venography is considered the gold standard for the detection of DVT in patients with stasis leg ulcers. However, venography is an invasive procedure that requires the use of radiation and contrast agents. CT venography may be useful for the exclusion of pulmonary embolism in patients with signs of deep venous thrombosis in the legs; however, CT venography still requires the injection of contrast media in the morbid limb to achieve optimal venous imaging of the extremities, which is dangerous for the limb [12]. Although time-of-flight MRV (TOF-MRV) is less invasive than conventional venography and CT venography, CT avoids side effects associated with iodinated contrast agents, such as renal damage, and is less operator-dependent than US [13,14,15]. Although MR angiography (MRA) are expensive and are hard to achieve on severely ill patients, it still had several advantages. Multimodal MRA techniques for reconstructing vascular structures mainly include (1) time-of-flight (TOF), (2) phase-contrast, and (3) ECG-gated turbo-spin-echo MRA. TOF-MRA was applied in evaluating arterial pathology such as atherosclerosis in 1998 [16]. Signal loss in a patent vessel from the saturation of in-plane flow is one of the most common pitfalls in two-dimensional (2D) TOF MR. The other disadvantage of TOF-MRV is that the FOV is small for each image obtained and that it requires considerable time to obtain a whole image of the lower extremity [17,18,19]. MRI with gadolinium-based contrast media is a relatively rapid method for imaging the lower extremities [20,21]. Although MRI does not involve radiation exposure, the noniodinated contrast agents involved in the imaging process still have undesirable effects. For example, nephrogenic sclerosing fibrosis (NSF) is a dangerous complication of gadolinium-based contrast agents in patients with pre-existing impairment of kidney function, and it may even occur in patients with normal renal function [22,23]. Phase contrast-MRI (PC-MRI) depends on phase shifts caused by blood flow. Thus, this technique permits the use of coronal or sagittal slice orientations with a field of view along the direction of the vessel of interest and can quantitatively measure the dynamics flow of the chosen region of interest. Prior studies applied PC-MRA for evaluating central nervous system pathology including vascular disease and hydrocephalus [24,25].

Traditional contrastless MRA, such as TOF-MRA and PC-MR angiogram, remain very time-consuming for imaging the whole lower extremity venous structures. The ECG-gated, multi-step turbo-spin echo technique (or Triggered angiography non-contrast-enhanced sequence magnetic resonance imaging, TRANCE MRI) offers the possibility of imaging the whole lower extremity vascular structures in clinical appliance. ECG-gating helps to adapt imaging times to different flow characteristics and, therefore, to optimize image quality faster. Although there is some related literature on non-contrast-enhanced MRA, most of them are associated with researchers using this technique to evaluate arterial diseases [26,27,28,29]. Our team has innovated the use of TRANCE-MRI for providing more valuable information for management of complicated lower venous diseases [5,6].

In this study, TRANCE-MRI revealed DVT in three patients, which could not be identified by Doppler US (Table 2). Congenital venous anomalies, possible MT syndrome, and pelvic congestion were detected by TRANCE-MRI only. Coexisting peripheral arterial disease was also clearly revealed by TRANCE-MRI of the arterial system, even in patients with renal insufficiency. Accordingly, this technique is valuable because it can help physicians avoid compressive therapy in limbs with arterial occlusive diseases [6,26]. Notably, in this study, TRANCE-MRI demonstrated that 21.7% of patients with stasis leg ulcers had no venous lesions (three patients with lymphedema and two patients with vasculitis); oral anticoagulation or venous interventions may thus be harmful to such patients. As for detecting prominent greater saphenous venous insufficiency, an indicator for GSV truncal ablation, the Interrater reliability of Cohen’s Kappa coefficient between TRANCE-MRI and venous duplex was 0.7315, which indicated that there is substantial agreement between the two imaging modalities.

In summary, the advantages of TRANCE-MRI for managing stasis leg ulcers are outlined as follows. First, TRANCE-MRI can outline arteries and veins from the pelvis to the toes in one examination and is valuable for detecting occult arterial diseases in stasis ulcers. Second, TRANCE-MRI does not require contrast medium injection or radiation. Therefore, it is particularly suitable for young patients; those with a history of allergy to contrast media; and those with chronic renal insufficiency, diabetes, or other comorbidities. Third, the perforating vein in the wounded ankle and calf can be clearly identified in TRANCE-MRI. Fourth, TRANCE-MRI can exclude most venous pathologies from the pelvis to the calf, thus enabling the precise delivery of oral anticoagulants.

We also determined a potential value of TRANCE-MRI in managing leg venous diseases: hemodynamic evaluation. We observed that the mean velocity and stroke volume in the GSV were paradoxically higher than those in the popliteal vein in patients with stasis leg ulcers with valvular insufficiency, confirmed by US. This implies that TRANCE-MRI may surpass existing diagnostic modalities, expanding from anatomic outlines to hemodynamic information.

However, we did identify some weaknesses of TRANCE-MRI in this study. First, TRANCE-MRI of the venous system may be laborious in the left iliac vessels, possibly because of the complex anatomy of the vessels as well as overlapping of vessels with different blood flow directions. The risk of inaccurate diagnoses may be reduced by factors and approaches such as increased diameter and number of collateral veins, constant filling defects, and intravascular US. Second, the TRANCE-MRI protocol requires 40 min for imaging (25 min for MRV and 15 min for MRA). Thus, it is not suitable for critical or irritable patients. Finally, TRANCE-MRI is expensive and not yet readily available at some institutions (Figure 4).

### Study Limitations

The major limitations of this study are its nonrandomized design and small sample size. We did not complete the sonographic evaluation in standing patients, which makes a good comparison between TRANCE MRI and sonography impossible. This study also did not adequately validate TRANCE-MRI against other imaging modalities in pelvis (such as intravascular ultrasound and CT venogram). Nonetheless, we identified some of the advantages and disadvantages of using TRANCE-MRI for the diagnosis and evaluation of stasis leg ulcers.

## 5. Conclusions

In this study, TRANCE-MRI revealed that five patients (21.7%) with stasis leg ulcers had no venous lesions. TRANCE MRI has good Interrater reliability between the Duplex study in the greater saphenous venous insufficiency with Cohen’s Kappa coefficient of 0.7315. It also potentially surpasses existing diagnostic modalities in terms of distinguishable hemodynamic figures. TRANCE-MRI is currently a safe and useful tool for examining stasis leg ulcers.

## Figures and Tables

**Figure 1 diagnostics-10-00707-f001:**
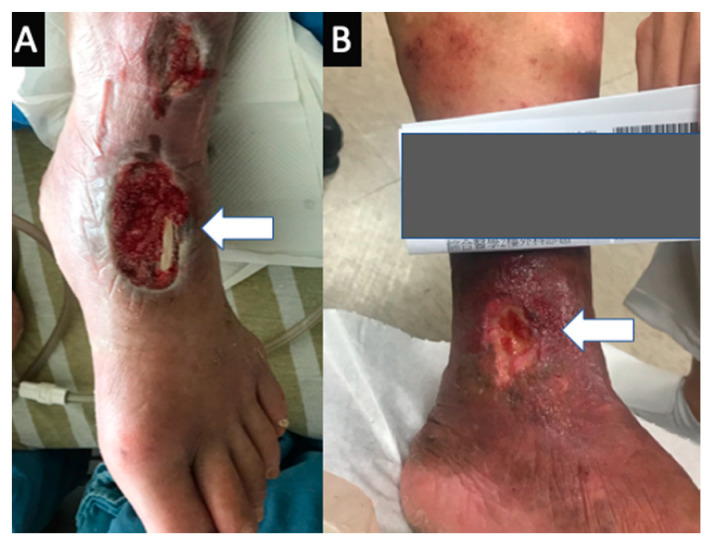
Typical stasis leg ulcer in the gaiter area. (**A**) Wet and nonhealing stasis leg ulcer with tendon exposure. This patient has no co-existed arterial nor connective tissue disorder. (**B**) Stasis leg ulcer with skin discoloration and granulation at the ankle. Typical stasis leg ulcer in the gaiter area.

**Figure 2 diagnostics-10-00707-f002:**
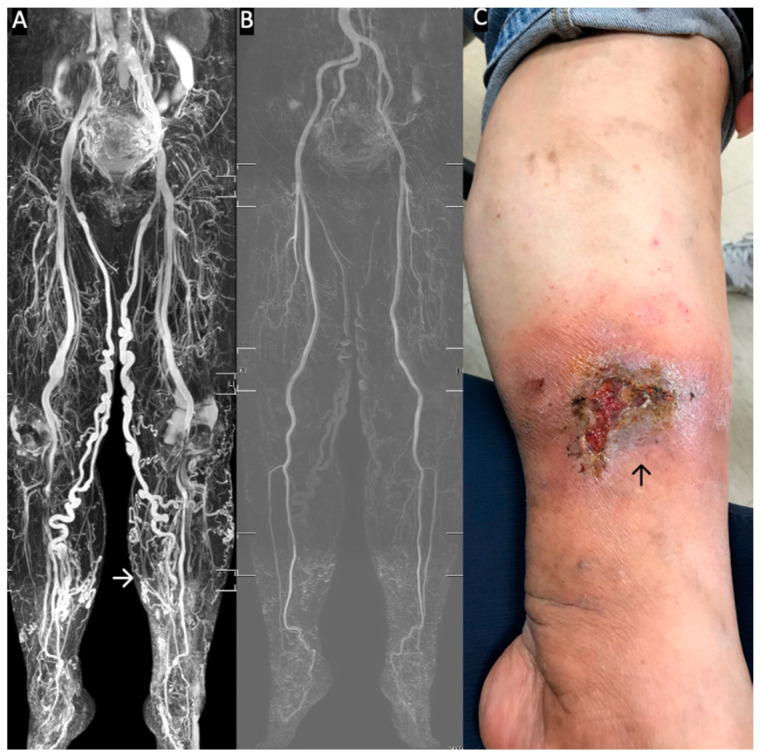
Standard Triggered angiography non-contrast-enhanced (TRANCE)-magnetic resonance imaging (MRI) evaluation for patients with stasis leg ulcers. (**A**) Full TRANCE-MRI. The TRANCE-MR image of the venous system clearly shows the tortious great saphenous vein (GSV) in both legs. The white arrow indicates the stasis leg ulcer. (**B**) The TRANCE-MRI can also identify the arterial system. There is no need for radiation or injection of contrast medium. (**C**) Stasis leg ulcer of the gaiter area. This wound healed one month later, after truncal ablation of the left GSV.

**Figure 3 diagnostics-10-00707-f003:**
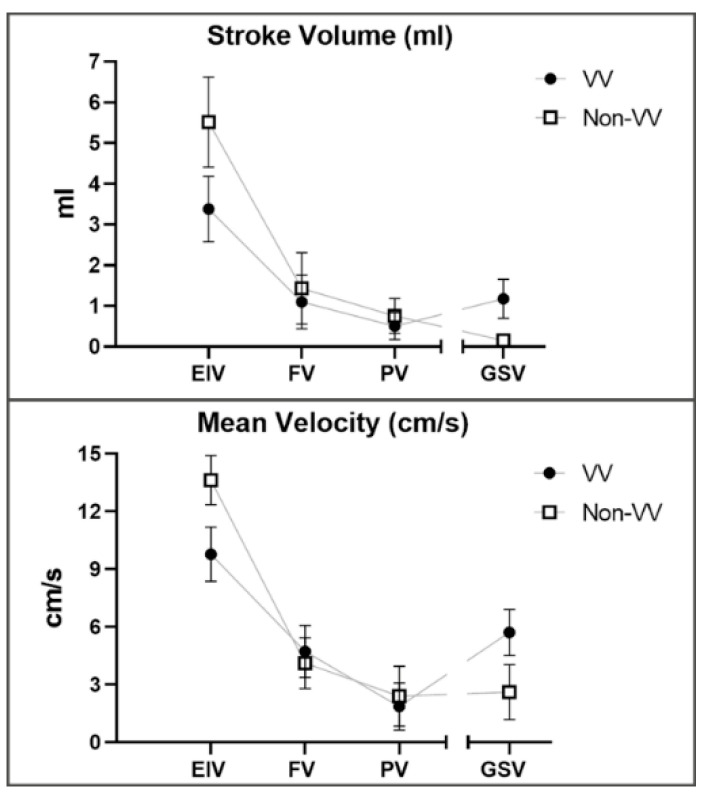
TRANCE-MRI hemodynamic parameters, consisting of stroke volume, forward flow volume, backward flow volume, regurgitant fraction, absolute volume, mean flux, stroke distance, and mean velocity, were analyzed in three valvular insufficiency cases and three venous occlusion cases. All patients with valvular insufficiency had paradoxically higher mean velocity and stroke volume in the GSV compared with the popliteal vein. EIV: external iliac vein; FV: femoral vein; PV: popliteal vein; GSV: greater saphenous vein; VV: venous valvular insufficiency; Non-VV: non venous valvular insufficiency.

**Figure 4 diagnostics-10-00707-f004:**
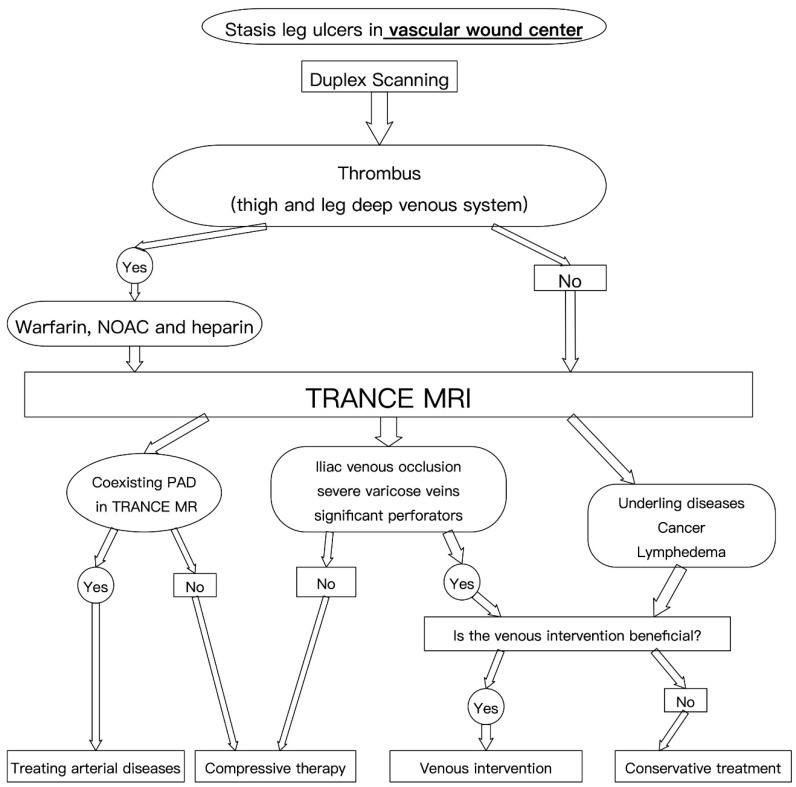
Therapeutic algorithm in the vascular wound center for stasis leg ulcers. NOAC: non-warfarin oral anticoagulation; PAD: peripheral arterial disease.

**Table 1 diagnostics-10-00707-t001:** Twenty-three patients receiving TRANCE-MRI for static ulcers in their lower extremities.

Title	Title
Male Gender (%)	18 (78.2%)
Age (years old)	61.47 ± 13.6
Substance use	
Smoking	3 (13%)
Alcohol	1 (4.3%)
Areca nuts	2 (8.6%)
Comorbidities	
Hypertension	10 (43.4%)
Diabetes Mellitus	6 (26.1%)
CAD	1 (4.3%)
Stroke	0 (0%)
Cancer	4 (18.1%)
Previous deep vein thrombosis	4 (18.1%)
Chronic Renal insufficiency	1 (4.3%)
Kidney transplantation	1 (4.3%)
Protein C, Protein S deficiency	2 (8.6%)
Factor V Leiden mutation	0 (0%)
Crushing limb with reconstruction	3 (13%)
Previous venous surgery	
Stripping	1 (4.3%)
GSV Ablation	2 (8.6%)
sclerotherapy or phlebectomy	1 (4.3%)
Venous intervention after TRANCE MR	
Truncal ablation of GSV	3 (13%)
Venous Angioplasty	2 (8.6%)
Compressive therapy	12 (52.2%)
Oral anticoagulation	14 (60.9%)

CAD: coronary arterial disease; GSV: greater saphenous vein; IVC: inferior vena cava.

**Table 2 diagnostics-10-00707-t002:** Comparison between TRANCE MRI and Duplex ultrasound scan.

	TRANCE MR for Venous System	Dupplex Ultrasound Scan
Saphenous-femoral venous junction reflux (ultrasound) or GSV/PV > 1 (TRANCE MRI)	9	9
Positive for thigh and calf deep venous thrombosis	6	3
Double inferior vena cava	1	0
May-Thurner syndrome	4	0
Pelvic congestion	2	0
Peripheral arterial occlusion	3	2
Negative for venous lesion	5	7

GSV: greater saphenous vein; PV: popliteal vein.

**Table 3 diagnostics-10-00707-t003:** Static leg ulcers: wound status group by TRANCE-MRI classification.

	Prominent Varicose Vein Favor Valvular Insufficiency	Venous Occlusion	No Venous Lesion	
Total number	9	9	5	
Wound Location				NS
Gaiter	9	8	5	
Proximal calf	1	4	0	
Foot and sole	0	1	1	
Number				NS
Single	8	4	2	
More than 1	1	5	2	
Depth				NS
over 5 mm	2	3	3	
Tendon exposure	0	0	1	
Recurrence	7	8	5	NS
Both legs	1	4	1	NS
Intervention				
Truncal ablation and stripping	2	2	0	NS
Venous angioplasty	0	2	0	NS
Compressive therapy	5	6	1	NS
Vacuum wound closure device	1	1	0	NS
oral anticoagulation	2	9	3	*p* = 0.002

NS: non-significant.

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
