# Peer review of "Stasis Leg Ulcers: Venous System Revises by Triggered Angiography Non-Contrast-Enhanced Sequence Magnetic Resonance Imaging"

_diagnostics, 2020, doi:10.3390/diagnostics10090707_

Round 1

Reviewer 1 Report

The Authors modified the manuscript according to most (not all) comments from reviewers. For example: leg skin ulcers due to vasculitis cannot be included between "stasis" ulcer.

Author Response

Dear reviewer 1

Point 1:

 For example: leg skin ulcers due to vasculitis cannot be included between "stasis" ulcer.

Response 1

Thanks for your warm remind.

We revised the paragraph as “Two patients were found with coexisted vasculitis” in this version. (line 146)

Reviewer 2 Report

This manuscript is ver interesting.

I think it is very useful for daily clinical practice.

Author Response

Dear Reviewer 2

Thanks for your comment and encourage.

Reviewer 3 Report

There are several problems that make this paper not suitable for publication.

  1. The Authors use the term “stasis leg ulcer”. This is an old term that is no longer used. The proper term is “venous leg ulcer” and it should be distinguished from other chronic wounds in the gaiter area. For example, the ulcer in Fig.1A does not look like a venous ulcer but rather a rheumatoid or a mixed arterio-venous ulceration, still no information is provided regarding this patient.
  2. All sonographic examinations were performed in the supine body position, which is against current recommendations on sonographic evaluation of low leg veins; only the assessment of patency of deep veins should be performed in the supine position, while reflux, especially in the superficial veins, should always be assessed in standing patients. Moreover, iliac and other pelvis veins were not assessed sonographically in this study, which makes a comparison between MRI and sonography impossible.
  3. Patients’ selection is unclear. No inclusion criteria are provided.
  4. There are also many misconcepts regarding pathophysiology of venous leg ulcers in this manuscript.
  5. In conclusion, this particular MRI protocol could be an interesting method of the diagnostics in venous leg ulcer patients, particularly regarding an assessment of the iliac veins. Still, the Authors should clearly show the advantage of this diagnostic method in comparison with standard ultrasonography, and also in comparison with IVUS, which is currently regarded the golden standard for the assessment of iliac veins.

Author Response

Reviewer 3

Thank you for reviewing our manuscript and giving us many instructive comments. Hope our effort could make this article more informative.

Point 1:

Authors use the term “stasis leg ulcer”. This is an old term that is no longer used. The proper term is “venous leg ulcer” and it should be distinguished from other chronic wounds in the gaiter area. For example, the ulcer in Fig.1A does not look like a venous ulcer but rather a rheumatoid or a mixed arterio-venous ulceration, still no information is provided regarding this patient.

Response 1

We adjust the nomination by additional description according to your comment.

The other reviewer (Reviewer 1) favors the use of “stasis leg ulcers”, to simulate possibly clinical scenario before the definite pathogenesis found.

We also add relevant information in Figure 1A

Change:

Back ground

Line 56 Stasis leg ulcers (or venous leg ulcer, formally)

Line 89 add “This patient has no co-existed arterial nor connective tissue disorder.

Point 2:

All sonographic examinations were performed in the supine body position, which is against current recommendations on sonographic evaluation of low leg veins; only the assessment of patency of deep veins should be performed in the supine position, while reflux, especially in the superficial veins, should always be assessed in standing patients. Moreover, iliac and other pelvis veins were not assessed sonographically in this study, which makes a comparison between MRI and sonography impossible.

Response 2

Thanks for your comment.

Most of those patients have wounded legs, thus they could not receive detailed sonograph survey in standing. Our CV echo team did not provide sonographic service for pelvic study. We add the information into the “study limitations”

Change:

Study limitations

Line 254

Meanwhile, we did not complete the sonographic evaluation in standing patients and no pelvic survey by echo, which makes a good comparison between TRANCE MRI and sonography impossible.

Point 3:

Patients’ selection is unclear. No inclusion criteria are provided.

Response 3

We revise the inclusion criteria of this study (TRANCE MRI for venous leg ulcers) according to your comment.

Change

All patients who suspected of having complex venous pathology in their lower extremities were included initially. Patients were excluded if they were pregnant or had non-MRI-compatible ferromagnetic implants. In addition, patients were excluded if they exhibited poor compliance or had multiple comorbidities that prevented them from lying down for the 1-hour TRANCE-MRI protocol. Initially, 104 patients were evaluated. One patient was excluded because of possible pregnancy, and three additional patients were either morbidly obese or restless, rendering them unable to continue with the MRI study. Among the remaining 100 patients, 23 received further survey for their stasis leg ulcers in the Wound and Vascular Center at our institution.

Point 4:

There are also many misconcepts regarding pathophysiology of venous leg ulcers in this manuscript.

Response 4

There are still many controversies in the pathophysiology of venous leg ulcer, thus we reviewed relevant references and cited in this version for the readers who need further refresh. (discussion, line 76-78) 

Point 5:

In conclusion, this particular MRI protocol could be an interesting method of the diagnostics in venous leg ulcer patients, particularly regarding an assessment of the iliac veins. Still, the Authors should clearly show the advantage of this diagnostic method in comparison with standard ultrasonography, and also in comparison with IVUS, which is currently regarded the golden standard for the assessment of iliac veins.

Response 5

We add this in the “study limitation” . We will try to include those modalities in the further study.

Change

Study limitation, line 257-258

This study also did not adequately validate TRANCE-MRI against other imaging modalities in pelvis (such as intravascular ultrasound and CT venogram)

This manuscript is a resubmission of an earlier submission. The following is a list of the peer review reports and author responses from that submission.

Round 1

Reviewer 1 Report

The Authors report results obtained by triggered non-contrast-enhanced MRI (TRANCE MRI) in legs with skin ulcerations.

In vascular medicine the most used term is “stasis ulcer”, whereas “static ulcer” could be misinterpreted with an ulcer that does not heal nor worses.

The Authors have recently published in BMC and in another journal (Diagnostic, Basel) the main findings obtained by TRACER-MRI in legs with venous disorders. There are no innovative concepts regarding TRACE-MRI of venous disorders with respect of their recent publications.Moreover, it is missing a historical review of the other techniques of contrastless MR venography.

Moreover, the pathophysiology of venous return and of venous ulcer development is outlined without considering current theories. Same statements about venous pathology (i.e., lines 133-134) and diagnostics (i.e., lines 170-172) do not correlate with current thought.

The Authors consider TRACE-MRI a tool for “treating” static leg ulcers (lines 200, 208-212). TRACE-MRI does not “treat” venous diseases.

The clinical evaluation of the legs included in this study is poorly reported: it is mandatory to know the (hypothesis regarding the) pathophysiology  of the 5 ulcerated legs without evident venous disorders. Same statements are misleading (i.e., lines 132-3)

The Authors report results obtained by triggered non-contrast-enhanced MRI (TRANCE MRI) in legs with skin ulcerations.

In vascular medicine the most used term is “stasis ulcer”, whereas “static ulcer” could be misinterpreted with an ulcer that does not heal nor worses.

The Authors have recently published in BMC and in another journal (Diagnostic, Basel) the main findings obtained by TRACER-MRI in legs with venous disorders. There are no innovative concepts regarding TRACE-MRI of venous disorders with respect of their recent publications.Moreover, it is missing a historical review of the other techniques of contrastless MR venography.

Moreover, the pathophysiology of venous return and of venous ulcer development is outlined without considering current theories. Same statements about venous pathology (i.e., lines 133-134) and diagnostics (i.e., lines 170-172) do not correlate with current thought.

The Authors consider TRACE-MRI a tool for “treating” static leg ulcers (lines 200, 208-212). TRACE-MRI does not “treat” venous diseases.

The clinical evaluation of the legs included in this study is poorly reported: it is mandatory to know the (hypothesis regarding the) pathophysiology  of the 5 ulcerated legs without evident venous disorders. Same statements are misleading (i.e., lines 132-3)

The Authors report results obtained by triggered non-contrast-enhanced MRI (TRANCE MRI) in legs with skin ulcerations.

In vascular medicine the most used term is “stasis ulcer”, whereas “static ulcer” could be misinterpreted with an ulcer that does not heal nor worses.

The Authors have recently published in BMC and in another journal (Diagnostic, Basel) the main findings obtained by TRACER-MRI in legs with venous disorders. There are no innovative concepts regarding TRACE-MRI of venous disorders with respect of their recent publications.Moreover, it is missing a historical review of the other techniques of contrastless MR venography.

Moreover, the pathophysiology of venous return and of venous ulcer development is outlined without considering current theories. Same statements about venous pathology (i.e., lines 133-134) and diagnostics (i.e., lines 170-172) do not correlate with current thought.

The Authors consider TRACE-MRI a tool for “treating” static leg ulcers (lines 200, 208-212). TRACE-MRI does not “treat” venous diseases.

The clinical evaluation of the legs included in this study is poorly reported: it is mandatory to know the (hypothesis regarding the) pathophysiology  of the 5 ulcerated legs without evident venous disorders. Same statements are misleading (i.e., lines 132-3)

Reviewer 2 Report

GENERAL COMMENT

This is an observational study on 23 patients enrolled over a 3-year period; no control group is available; data analysis is conducted retrospectively.

The study aims to demonstrate that TRANCE MRI can be a safe and reliable technique useful in the diagnosis and management of venous leg ulcers in clinical practice.

The method does not include the comparison of TRANCE MRI with the gold standard reference technique represented by conventional venography, but only with Doppler ultrasound.

Furthermore, the authors do not mention or provide bibliographic data on sensitivity, specificity, positive predictive value, negative predictive value and accuracy of TRANCE MRI.

The sample size is very small.

Only 5 females were enrolled, although female sex is known to be one of the risk factors for developing venous leg ulcers.

The study of hemodynamic parameters is performed on two even smaller subsamples (3 out of 9 patients with venous insufficiency and 3 out of 6 patients with venous thrombosis).

Therefore, the study design and the data presented do not allow to draw reliable conclusions regarding the purpose of the study.

SPECIAL COMMENTS

A thorough revision of the English language is required.

Discuss the results of the study and not only the theoretical advantages and disadvantages of TRANCE MRI compared to other diagnostic techniques not used in the study.

Provide conclusions relevant to the study results rather than generic.

Table 1 needs to be improved: the authors should provide complete clinical data, including the presence in the study population of the most common risk factors for the development of venous leg ulcers, such as obesity, trauma, immobility, previous deep vein thrombosis, previous phlebitis, congenital situations such as factor V Leiden mutation or congenital absence of veins.

Figure 3 is unclear and should be improved; provide a list of abbreviations.

Focus on all the aspects discussed above and improve the study's design, numbers and data to make it more powerful than it is now, in order to respond to the important topic you are studying on.
